# Single-Crystalline Metal Oxide Nanostructures Synthesized by Plasma-Enhanced Thermal Oxidation

**DOI:** 10.3390/nano9101405

**Published:** 2019-10-02

**Authors:** Bin Guo, Martin Košiček, Junchi Fu, Yazhou Qu, Guanhua Lin, Oleg Baranov, Janez Zavašnik, Qijin Cheng, Kostya (Ken) Ostrikov, Uroš Cvelbar

**Affiliations:** 1School of Electronic Science and Engineering, College of Energy, Xiamen University, Xiamen 361005, China; guobin19920917@163.com (B.G.); fjc@ronbaymat.com (J.F.); qyz1991@live.com (Y.Q.); guanhualin1992@gmail.com (G.L.); 2Shenzhen Research Institute of Xiamen University, Shenzhen 518000, China; 3Jožef Stefan Institute, Jamova Cesta 39, SI-1000 Ljubljana, Slovenia; martin.kosicek@ijs.si (M.K.); oleg.baranov@ijs.si (O.B.); janez.zavasnik@ijs.si (J.Z.); 4Jožef Stefan Postgraduate School, Jamova Cesta 39, SI-1000 Ljubljana, Slovenia; 5National Aerospace University, Kharkov 61070, Ukraine; 6School of Chemistry, Physics and Mechanical Engineering, Queensland University of Technology, Brisbane QLD 4000, Australia; kostya.ostrikov@qut.edu.au; 7Joint CSIRO-QUT Sustainable Processes and Devices Laboratory, Lindfield NSW 2070, Australia

**Keywords:** metal oxide nanostructures, plasma-enhanced thermal oxidation, growth mechanism, material characterization, nanostructure growth modelling

## Abstract

To unravel the influence of the temperature and plasma species on the growth of single-crystalline metal oxide nanostructures, zinc, iron, and copper foils were used as substrates for the study of nanostructure synthesis in the glow discharge of the mixture of oxygen and argon gases by a custom-made plasma-enhanced horizontal tube furnace deposition system. The morphology and microstructure of the resulting metal oxide nanomaterials were controlled by changing the reaction temperature from 300 to 600 °C. Experimentally, we confirmed that single-crystalline zinc oxide, copper oxide, and iron oxide nanostructures with tunable morphologies (including nanowires, nanobelts, etc.) can be successfully synthesized via such procedure. A plausible growth mechanism for the synthesis of metal oxide nanostructures under the plasma-based process is proposed and supported by the nanostructure growth modelling. The results of this work are generic, confirmed on three different types of materials, and can be applied for the synthesis of a broader range of metal oxide nanostructures.

## 1. Introduction

One-dimensional nanostructures have unique mechanical, optical, and electronic properties and represent an important platform for a wide range of applications [1,2,3]. Especially, one-dimensional metal oxide nanomaterials show great potential for nano-electronic applications due to their magnetic and redox characteristics [4,5].

For our experiment, three different and most reported metal oxide systems were investigated, based on their unique characteristics, properties, and applications, as well as easy comparison with the existing results. In particular, ZnO is a metal-oxide (MO) semiconductor with a wide band gap of 3.37 eV and a large exciton binding energy of 60 meV [2,6]. Due to its unique properties, it is exploited in a wide range of applications, such as ultraviolet lasers, detectors, optical waveguides, laser diodes, surface acoustic wave devices, transparent electrode coatings, acousto-optic devices, thin film transistors, piezoelectric sensors, solar cells, etc. [7,8,9,10]. Fe_2_O_3_ is a transition metal oxide with a band gap of 2.2 eV, which has been used for centuries in the food industry, medicine, pigments, and ceramics [11,12,13]. Especially nanosized Fe_2_O_3_ exhibits catalytic, magnetic, and gas-sensitive properties that can be further exploited [14,15]. CuO is a *p*-type narrow bandgap semiconducting material with a band gap of 1.2 eV [16]. In particular, CuO nanomaterials have optical, electrical, magnetic, and catalytic properties, which have great application potential as catalytic, photothermal photoconductive materials, field emitters, gas sensors, etc. [17,18,19].

Currently, the most commonly used synthesis routes for the above-mentioned metal oxide nanomaterials include template method [20], hydro- or solvothermal synthesis [21], laser ablation method [22], thermal oxidation method [23], etc. Important criteria for large-scale synthesis of MO nanomaterials are their production cost, quality and uniformity of such obtained nanomaterials, and controllable morphology of nanostructures. The template method can assure good control over the morphology of nanostructures, but it is expensive and time-consuming [24]. Hydrothermal synthesis is fast, efficient and relatively cheap, but the morphology of the product is only poorly controllable by, for example, using surfactants [24]. Laser ablation synthesis can provide more control over the morphology and quality of the nanomaterials, but the process itself is expensive and unsuitable for mass production [25]. Thermal oxidation method can provide good control over the morphology of the nanomaterials, but the reproducibility of the preparation process is insufficient [26,27,28,29].

Recently, plasma-assisted synthesis of various nanostructures has been extensively studied [30,31]. For the synthesis of MO nanostructures, metal substrates were treated with oxygen or other gases under plasmas [32]. One of the most significant advantages of such an approach is the short synthesis time, which is reduced by an order of magnitude compared to thermal methods. To form an oxide via thermal oxidation synthesis process, an oxygen molecule has to adsorb on the heated substrate, where it dissociates to form atomic oxygen, and subsequently, the atomic oxygen reacts with metal [33]. On the contrary, plasma already contains atomic oxygen species; therefore, by using oxygen plasmas as a treating agent, one can skip the oxygen dissociation step [34]. Methods utilizing plasmas are cheap, easy to scale-up, and are environmentally friendly, which makes them perfect for the large-scale or even industrial synthesis of nanostructures. Up to now, multiple routes of the plasma synthesis have been described, utilizing different mechanisms of nanostructure growth and control [31].

For the Zn–Fe–Cu oxide model system investigated in this paper, a few different plasma-assisted synthesis approaches have been proposed [30]. Namely, a promising route to obtain ZnO nanostructures in bulk quantities within a very short time is the so-called flight through method [30], where Zn particles enter plasma on one side, fly through (usually by free fall), and exit as oxide nanowires. Through the plasma synthesis method on the Zn or Zn/Cu surface, ultra-thin zinc oxide nanowires (NWs) can be obtained as well [35]. Similarly, plasma methods have been used for the synthesis of Cu oxide [36,37,38,39,40] and Fe oxide nanostructures [30,31,41]. Since copper and iron have a relatively high melting point (1085 and 1538 °C, respectively) compared to Zn (420 °C), the growth mechanism of metal oxide nanostructures, although following the same concept, slightly differs from that of zinc oxide nanostructures [31].

Two basic growth mechanisms of MO nanostructures are proposed as methods of direct exposure of a metal substrate to the plasma [30,37], namely solid–liquid–solid (SLS) and solid–solid (SS) growth mechanisms. In the SLS growth mechanism, tiny droplets of liquid metal form on the solid metal surface when the metal is exposed to temperatures close to its melting point. Oxygen atoms from the plasma first diffuse in the liquid phase of the metal and form solid oxides, which then act as nucleation sites for further nanowire growth. This mechanism is common in metals with low melting points [37]. On the other hand, in the SS growth mechanism, oxygen atoms pass directly into solid metal phase [37]. Oxygen atoms are incorporated inside the metal phase on hotter sites and metal oxide phases are formed, at which point an oxide layer is grown and nuclei act as sites for further nanowire growth [31].

In this work, a custom-made plasma-enhanced horizontal tube furnace deposition system, i.e., plasma-enhanced thermal oxidation (PETO), has been developed and described in an attempt to synthesize a broad range of metal oxide nanostructures, including ZnO, Fe_2_O_3_, CuO, etc. Using the PETO method, we were able to produce stable and high-quality metal oxide nanomaterials at a low processing temperature and within a short growth time. Moreover, we have systematically investigated the structural and morphological properties of the synthesized low-dimensional MO nanostructures. Finally, a viable growth mechanism supported by the nanostructure growth modeling has been proposed to interpret the obtained experimental results. 

## 2. Materials and Methods

Zinc, iron, and copper foils were used as substrates for the preparation of corresponding metal oxides nanostructures by custom-made plasma-enhanced horizontal tube furnace deposition system. The system is composed of a horizontal quartz-glass one-temperature zone tube furnace, an RF power generator, a vacuum system, a heating device, and a gas input system, as shown in Figure 1. Two cylindrical metal electrodes with sharp tips were placed at both ends of the quartz tube to produce plasmas. The size of the metal foils used in the experiment was 10 mm × 10 mm × 0.25 mm. At first, metal foils were ultrasonically cleaned with deionized water, acetone, and absolute ethanol for 20 min, respectively, and then placed into the reactor. The entire system was evacuated to a base pressure of 10 Pa or less. The quartz tube was first heated up to working temperature (300–600 °C) within 20 min, and then a gas mixture of argon and oxygen was introduced. Flow rates of Ar and O_2_ gases were set as 45 and 5 sccm, respectively. Thereafter, we turned on the plasma power and adjusted the current to 1.3 A. For all the samples, plasma was discharged for 30 min. Finally, the plasma power was turned off, the gas inlets were cut off, and the system was allowed to cool down to room temperature.

The crystal structure of the synthesized metal oxide nanostructures was determined by X-ray diffraction (XRD, Ultima IV, Rigaku, Tokyo, Japan) operating in a locked couple mode, wherein the incident X-ray wavelength was 1.54 Å (Cu K_α_ line) at 40 kV and 40 mA. The microstructure of the synthesized metal oxide nanostructures was characterized by field emission scanning electron microscope (SEM, Supra 55, Carl Zeiss, Oberkochen, Germany) operated at 5 kV and high-resolution transmission electron microscope (TEM, JEM 2100, Jeol, Tokyo, Japan) operated at 200 kV. Raman measurements were conducted via Raman spectrometer (Xplora Plus, Horiba, Kyoto, Japan) using a 638 nm semiconductor laser for excitation. Before the measurement, Raman spectrometer was calibrated with single crystal silicon. The Raman spectra were recorded with a power of 1 mW and an integration time of 30 s.

## 3. Results and Discussion

### 3.1. XRD Analysis

The phase composition of the synthesized MO nanostructures on Zn, Cu, and Fe foils was first identified by XRD. The results are summarized in Figure 2a–c, presenting the XRD spectra of the synthesized zinc oxide, copper oxide, and iron oxide nanostructures at different growth temperatures, respectively. As shown in Figure 2a, at a growth temperature of 500 °C, diffraction peaks corresponding to ZnO (JCDPS card no. 65-3411) can be observed; these peaks are characteristic for a hexagonal wurtzite-type polymorph. In Figure 2b, characteristic diffraction peaks for CuO (JCDPS card no. 45-0937) can be observed at a growth temperature of 500 °C, while in Figure 2c, diffraction peaks, which can be identified for Fe_2_O_3_ (JCDPS card no. 36-1451), are observed at a high growth temperature of 600 °С. Also, the peaks of Zn (JCDPS card no. 65-3358), Cu (JCDPS card no. 04-0836) plus Cu_2_O (JCDPS card no. 05-0667), and Fe (JCDPS card no. 52-0513) plus Fe_3_O_4_ (JCDPS card no. 65-3107) appear in Figure 2a–c, respectively. With the increase of the growth temperature, the diffraction peaks of Zn, Cu, and Fe_3_O_4_ gradually become weaker and disappear almost completely, indicating the decomposition of the oxide nanostructures at elevated temperatures. Based on the XRD experiments, the optimum growth temperature window for each oxide phase has been determined. 

The results of XRD analysis confirm the successful synthesis of ZnO, CuO, and Fe_2_O_3_ by exposing zinc, copper, and iron foils to the glow discharge of the mixture of oxygen and argon gases by our custom-made plasma-enhanced horizontal tube furnace deposition system. 

### 3.2. Electron Microscopy Characterization

The influence of the growth temperature on the surface morphology of the Zn, Cu, and Fe oxide nanostructures are presented in Figure 3, Figure 4 and Figure 5, respectively. In the case of ZnO nanostructures (Figure 3), the NWs grown at 400 °C had a sturdy base with a sharp tip. The average diameter and length of these nanostructures were approximately 396 nm and 0.4 μm, respectively. With an increase of the growth temperature to 450 °C, a mixture of finer and longer NWs and nanobelts were obtained, with an average diameter and length about 110 nm and 1.4 μm, respectively. With further increase in the growth temperature to 500 °C, the average diameter and length of the nanostructures were about 130 nm and 1.8 μm, respectively. From these observations, we can derive the basic trend: with the increase of the growth temperature, ZnO nanostructures preferentially grow in c-axis (NWs become longer), while nanobelts eventually entirely disappear. From the TEM micrographs of ZnO NWs grown at a processing temperature of 450 °C and a deposition time of 0.5 h (Figure 3d,e), we could see well-defined single-crystal ZnO with a uniform diameter of 110 nm and a sharp tip. It is noteworthy that the length of the nanowire could not be precisely measured, possibly because of the effect of sonication during the sample preparation. Selected-area electron diffraction pattern (SAEDP) corresponds to the hexagonal wurtzite-type ZnO structure (Figure 3d-inset) and indicates that the ZnO NWs grow in the <0001> direction.

Plasma treatment of the copper foil at 300 °C (Figure 4a) resulted in sporadic Cu-oxide NWs grown on the surface of the copper foil. The average diameter and length were approximately 70 nm and 0.8 μm, respectively. With the increase of the growth temperature to 350 °C (Figure 4b), we were able to observe the increased density of the Cu-oxide NWs, although the average diameter and length (about 75 nm and 0.9 μm, respectively) did not increase significantly. With a further increase of the growth temperature to 400 °C (Figure 4c), the obtained copper oxide nanostructures became sturdy, and their average diameter and length were about 600 nm and 1.4 μm, respectively. TEM observations of Cu oxide NWs grown at a processing temperature of 300 °C and a deposition time of 30 min (Figure 4d,e) revealed a single-crystalline uniform elongated NW with a diameter of 70 nm and a length of 800 nm. The SAEDP (Figure 4d-inset) corresponds to monoclinic single-crystalline CuO and indicates that the CuO NW is elongated in the <111> direction.

During the plasma experiments with the Fe foil (Figure 5a–c), we observed no surface nanostructures at a growth temperature below 400 °C. When the growth temperature increased to 500 °C, some sparse iron oxide nanobelts could be observed (Figure 5b). Moreover, when the growth temperature was further increased to 600 °C, the whole surface of the Fe foil was covered with dense iron oxide nanobelts (Figure 5c). Some of them were scraped for further TEM analysis (Figure 5d–e), and initial observation (Figure 5d) revealed partially broken, but otherwise well-developed crystalline flattened NWs. SAEDP corresponds to hexagonal α-Fe_2_O_3_ (hematite). From the high-resolution TEM image (Figure 5e) and the corresponding SAEDP, we concluded that the preferred growth direction of the Fe_2_O_3_ NWs is along the [001] axis. 

SEM and TEM analysis revealed that ZnO, CuO, and Fe_2_O_3_ nanostructures with the morphology of nanowire and nanobelt can be obtained in our process. Moreover, the diameter and length of the synthesized ZnO, CuO, and Fe_2_O_3_ nanostructures can be tailored via the change of growth temperature. Table 1 summarizes the characteristics of zinc oxide, copper oxide, and iron oxide nanostructures obtained in this work.

### 3.3. Raman Analysis

The synthesized ZnO nanostructures had a hexagonal (wurtzite-type) structure (SG C_6v_). In this structure, there are 12 normal vibration modes of ZnO (three of 12 modes are acoustic phonon vibrations, while nine of 12 modes are optical phonon vibrations). The normal lattice vibrations in the cluster basis theory satisfy Γ_opt_ = A_1_(*z*) + 2B_1_ + E_1_(*x*,*y*) + 2E_2_, where A_1_ contains A_1_(LO) and A_1_(TO), while E_1_ contains E_1_(LO) and E_1_(TO). These A_1_ and E1 optical phonons are frequently observed in the Raman and infrared spectroscopy. In addition, two B_1_ modes are inactive for Raman and infrared spectroscopy, and they are usually called silent mode. Also, two E_2_ modes are nonpolar optical phonon and are Raman active [42,43].

A typical Raman spectrum of ZnO nanostructures produced at a processing temperature of 450 °C is presented in Figure 6a; there is a strong Raman scattering peak located at 437 cm^−1^. This peak is the nonpolar E_2_ mode of the ZnO phase. A weak peak located at 559 cm^−1^ is the A_1_ (LO) vibration mode of the ZnO crystal [27]. This peak is specific for ZnO nanostructures, which is consistent with the SEM result. The peak of the E_1_ (LO) vibration mode of the ZnO crystal located at approximately 583 cm^−1^ was not found in the Raman spectrum, and this peak is related to the oxygen deficiency of zinc oxide. Therefore, it is reasonable to conclude that the synthesized ZnO nanomaterials contained few oxygen-related defects. 

The space group of copper oxide is C^6^_2h_. There are 12 optical phonon active modes, and these vibration modes satisfy Γ_opt_ = 4A_u_ + 5B_u_ + A_g_ + 2B_g_. Among these vibration modes, A_g_ and 2B_g_ are Raman active. According to the Raman spectrum of copper oxide nanostructures produced at a processing temperature of 300 °C shown in Figure 6b, there are three Raman scattering peaks located at 295, 343, and 630 cm^−1^ in the range of 200–700 cm^−1^, which represent vibration modes for A_g_, B_1g_ and B_2g_, respectively. These three peaks are red-shifted compared to the corresponding Raman peaks of the copper oxide crystals located at 303, 350, and 636 cm^−1^, respectively, which is attributed to the tensile stress [43].

The Raman spectrum of the iron oxide nanostructures produced at a processing temperature of 600 °C is presented in Figure 6c. Fe_2_O_3_ belongs to the D^6^_3d_ space group. Six Raman peaks located at 226, 244, 293, 410, 501, and 613 cm^−1^ in the range of 100–700 cm^−1^ are attributed to the iron oxide nanostructures. The Raman peaks located at 226 and 244 cm^−1^ originate from A_1g_ Raman vibration modes while the other four peaks located at 293, 410, 501, and 613 cm^−1^ originate from E_g_ Raman vibration modes. In addition, the peak located at 659 cm^−1^ belongs to Fe_3_O_4_ [15].

Raman analysis further substantiated that ZnO, CuO, and Fe_2_O_3_ nanostructures can be obtained in our custom-made plasma-enhanced horizontal tube furnace deposition system.

### 3.4. Growth Mechanism

The present work is a catalyst-free plasma-assisted processing route for the production of single-crystalline metal oxide nanostructures. However, in the presence of the plasma species, classical growth mechanisms from thermal oxidation have to be modified. In the described synthesis, not only the surface temperature but also the plasma plays an essential role by exciting the gas reactants and thus promoting the chemical reaction. As shown in Figure 7, under the optimum processing temperature, small metal particles are formed on the surface of the metal foil. Then, in the O_2_-Ar gas atmosphere, oxygen gas is ionized into O^+^ and O^2+^ ions while argon gas assists oxygen gas to be ionized effectively under the plasma discharge. After that, oxygen molecules are also converted to metastable oxygen atoms, neutral oxygen atoms, excited oxygen molecules, molecular oxygen ions, etc. Finally, these oxygen species react with metal atoms on the surface, and subsequently, metal oxide nanostructures are formed. Some of the chemical reactions occurring during the growth process for our system are described below:O^+^ + Ar → O* +Ar^+^(1)
O^+^ +O_2_ → O* + O_2_^+^(2)
O* +Zn → ZnO
O* +Cu → CuO
O* +Fe → Fe_2_O_3_.(3)

In the above Equations (1)–(3), O* represents an activated or metastable oxygen atom [31,44].

It is worthwhile to mention that the surface of the metal particles can form micro-hillocks due to the surface roughness of the metal foils during Ar ion bombardment. The surface energy of these micro-hillocks is high so that the activated oxygen atoms can adsorb on these micro-hillocks and then the metal particles will gradually transform to low-dimensional metal oxide nanostructures. This is an entropy reduction process, i.e., Δ*S* < 0. Owing to the spontaneous process, Gibbs free energy (Δ*G*) is also less than zero. According to the equation ΔG=ΔH−TΔS, it can be concluded that the enthalpy is reduced, so this is an exothermic process [43,44]. Notably, the adsorption of metal oxide species on the substrate is an exothermal process. Also, the recombination of activated oxygen atoms in the gas environment, the adsorption of argon and oxygen ions on the substrate, and the reaction of oxygen ions with metal particles release some extra energy on the metal micro-hillocks [31,44,45]. 

When the experimentally measured length and diameter of CuO and ZnO nanowires are considered at different growth temperatures, the difference in the aspect ratios becomes apparent. Specifically, CuO nanowires, grown in the temperature range from 300 °C to 400 °C, were initially long and thin at 300 °C (aspect ratio is about 11.4), and gradually changed to the bulk shape with the aspect ratio of about 2.3 at 400 °C. At the same time, ZnO nanowires started to grow at 400 °C and they also had the bulk shape (aspect ratio of about 1.0), which then turned to an elongated structure with the aspect ratio of about 13.8 when the growth temperature rose to 500 °C. 

To explain the difference in the growth behavior, a theoretical model described elsewhere [38] was applied. For the modelling we selected CuO and ZnO nanostructures due to similarity of their chemical formula and similarity of the observed geometry of the growing nanostructures (as Fe_2_O_3_ nanostructures exhibit a more ribbon-like structure with pronounced anisotropy perpendicular to the vertical growth direction).

Table 2 lists the experimental and theoretical values for the ZnO and CuO nanostructures synthesized at different growth temperatures and a growth time of 30 min. One can notice the values *L_nw_* (30 min) and *D_nw_* (30 min) of ZnO and CuO nanostructures are very close to the corresponding experimentally measured values.

The experimental results are plotted in Figure 8, where the difference in growth behavior is clearly seen. The growth temperature for ZnO shifted to the higher temperature, and the length of the nanowires exhibited monotonic growth at the temperature increase. Qualitatively, this behavior can be explained by the increased diffusion of the species at the elevated temperature, yet the adsorption of oxygen on the surface is still significant. However, the diameter of the CuO nanostructures also increased with the temperature, while ZnO nanostructures showed the opposite trend. Our numerical simulations reveal the nature of the process.

When trying to fit the simulation with the experimental data, we found that two parameters were responsible for the difference in the growth behavior, namely, the activation energy *ε_d_* of the copper (or zinc) diffusion and the density of the oxygen molecules adsorbed on the substrate surface.

In the model, the copper (or zinc) diffusion is described with the expression:(4)$$D_{a}=D_{0}exp\left(-\frac{e\varepsilon_{d}}{k_{B}T_{s}}\right),$$
where *ε_d_* is the activation energy, *eV*; *D*_0_ is a constant; *T_s_* is the growth temperature (K). The adsorption of oxygen molecules is expressed as a ratio of the surface density *n_O_*_2_ of the molecules (m^−2^) to the density *n*_0_ of the adsorption nodes (m^−2^) on the substrate surface, and is described by the Langmuir adsorption isotherm [46]:(5)$$\frac{n_{O2}}{n_{0}}=\frac{P_{O2}}{P_{0}+P_{O2}},$$
where *P_O_*_2_ is the partial pressure of oxygen, Pa; *P*_0_ is a constant that does not depend on the pressure *P_O_*_2_, but depends on the temperature; *n*_0_ is a surface density of the oxide adsorption nodes. The value of the constant *P*_0_ is expressed as [46]
(6)$$P_{0}={\left(\frac{M_{O2}}{2\pi h^{2}}\right)}^{3/2}{\left(k_{B}T_{s}\right)}^{5/2}exp\left(-\frac{e\varepsilon_{aO2}}{k_{B}T_{s}}\right),$$
where *M_O_*_2_ is the mass of the oxygen molecule, kg; *ε_aO_*_2_ is the oxygen adsorption energy, eV; *h* is Planck’s constant.

The partial pressure of oxygen is expressed as
(7)$$P_{O2}=\frac{\phi_{O2}}{\phi_{Add}+\phi_{O2}}P,$$
where *ϕ_O_*_2_ and *ϕ_Add_* are the flow rates of the oxygen and additional gas (argon, in our case) sccm and *P* is the total gas pressure in the chamber, Pa. 

We applied the different values of the diffusion activation energy *ε_d_* and oxygen adsorption energies *ε_aO_*_2_ to fit the experiment and obtained the growth dependencies of the nanowire length (*L_nw_*) and diameter (*D_nw_*), shown in Figure 9a for CuO and in Figure 9b for ZnO. 

Generally, we calculated the growth dynamics for three values of the growth temperature for both oxides: 300, 350, and 400 °C for CuO; 400, 450, and 500 °C for ZnO. However, the dependencies for the intermediate temperatures (350 °C for CuO and 450 °C for ZnO) almost superimposed with the characteristics for 300 °C for CuO and 500 °C for ZnO; and thus they are not shown in Figure 9 a–b. However, the diffusion activation energy *ε_d_* and oxygen adsorption energies *ε_aO_*_2_ used in the calculations were quite different, and they are present as dots in Figure 9c (*ε_d_*) and Figure 9d (*ε_aO_*_2_). While considering the energies, one can notice that the diffusion activation energy *ε_d_* for CuO seemed to saturate on the upper limit (1.7 eV), while the diffusion activation energy *ε_d_* for ZnO seemed to saturate on the lower limit (1.87 eV). It should be stressed that the energies were calculated for the plasma-enhanced growth and these adsorption and diffusion energies can differ significantly from the values predicted for the thermal growth on ideal crystal lattice due to the ion bombardment of the surface. Then, we speculated that the dependencies should have a similar behavior at the dependence on the temperature, which means that both energies should have saturated values. As a result, the analytical expressions were obtained as follows:(8)$$\varepsilon_{d\left(CuO\right)}=\left(1-\xi_{d\left(CuO\right)}\right)\varepsilon_{d\left(CuO\right)}^{low}+\xi_{d\left(CuO\right)}\varepsilon_{d\left(CuO\right)}^{high}$$
(9)$$\varepsilon_{d\left(ZnO\right)}=\left(1-\xi_{d\left(ZnO\right)}\right)\varepsilon_{d\left(ZnO\right)}^{low}+\xi_{d\left(ZnO\right)}\varepsilon_{d\left(ZnO\right)}^{high}$$
where $\varepsilon_{d\left(CuO\right)}^{low}=1.5$ eV; $\varepsilon_{d\left(CuO\right)}^{high}=$ 1.7 eV; $\varepsilon_{d\left(ZnO\right)}^{low}=1.87$ eV; $\varepsilon_{d\left(ZnO\right)}^{high}=$ 2.02 eV.

Similar arguments led us to the conclusion that the oxygen adsorption energies *ε_aO_*_2_ can be fit with the following equations
(10)$$\varepsilon_{aO2\left(CuO\right)}=\left(1-\xi_{aO2\left(CuO\right)}\right)\varepsilon_{aO2\left(CuO\right)}^{low}+\xi_{aO2\left(CuO\right)}\varepsilon_{aO2\left(CuO\right)}^{high}$$
(11)$$\varepsilon_{aO2\left(ZnO\right)}=\left(1-\xi_{aO2\left(ZnO\right)}\right)\varepsilon_{aO2\left(ZnO\right)}^{low}+\xi_{aO2\left(ZnO\right)}\varepsilon_{aO2\left(ZnO\right)}^{high}$$
where $\varepsilon_{aO2\left(CuO\right)}^{low}=1.2$ eV; $\varepsilon_{aO2\left(CuO\right)}^{high}=$ 1.7 eV; $\varepsilon_{aO2\left(ZnO\right)}^{low}=1.61$ eV; $\varepsilon_{aO2\left(ZnO\right)}^{high}=$ 1.9 eV.

In the above expressions, the functions $\xi_{i\left(j\right)}$ are described by the equation
(12)$$\xi_{i\left(j\right)}=\frac{exp\left[\frac{T_{s}-\left(L_{i\left(j\right)}-\delta L_{i\left(j\right)}\right)}{\delta L_{i\left(j\right)}}\right]}{1+exp\left[\frac{T_{s}-\left(L_{i\left(j\right)}-\delta L_{i\left(j\right)}\right)}{\delta L_{i\left(j\right)}}\right]}$$
with parameters $L_{d\left(CuO\right)}=$ 345 K, $\delta L_{d\left(CuO\right)}=$ 20 K; $L_{d\left(ZnO\right)}=$ 495 K, $\delta L_{d\left(ZnO\right)}=$ 20 K; $L_{aO2\left(CuO\right)}=$ 385 K, $\delta L_{aO2\left(CuO\right)}=$ 19 K; $L_{aO2\left(ZnO\right)}=$ 514 K, $\delta L_{aO2\left(ZnO\right)}=$ 24.5 K; and the function $1-\xi_{i\left(j\right)}$ has the shape of the Fermi–Dirac distribution [46].

The results of the approximation are shown in Figure 9c,d. From the analysis of the graphical dependencies, it can be seen that the slopes of the curves on both Figure 9c,d are almost the same, i.e., 2.2 meV/°C for diffusion (*ε_d_*), and 2.6 meV/°C for adsorption (*ε_aO_*_2_). This result allows us to assume that the nature of the change for the adsorption and diffusion energies on the growth temperature is the same and does not depend on the oxide chemical composition.

In addition, we calculated the dependence of the aspect ratio of the nanowires for CuO and ZnO, shown for different growth temperatures in Figure 9e. In spite of the fact that the experimentally measured values exhibited short and bulk nanowires for the growth temperature of 400 °C for both oxides, the calculations show the possibility to obtain the structure with a rather high aspect ratio of about 20 to 40 for the short plasma exposure time of about 1–2 min, yet the length of the nanowires were quite small. It should be noted that the possibility of growing the high-aspect nanowires over the short exposure time was confirmed by the experimental results obtained for the plasma-enhanced growth of CuO nanowires described in our previous paper [38].

Then, the calculated energies were used to estimate the relative probability of the nanowire growth described by the expression [47]:(13)$$P_{\mathrm{rel-growth}}=\frac{\frac{P_{O2}}{P_{0}\left(T_{s}\right)+P_{O2}}exp\left(-\frac{e\varepsilon_{d}}{k_{B}T_{s}}\right)}{\frac{P_{O2}}{P_{0\left(T_{s\left(max\right)}\right)}+P_{O2}}exp\left(-\frac{e\varepsilon_{d}}{k_{B}T_{s\left(max\right)}}\right)}$$
where *T*_*s*(*max*)_ is the temperature where the maximum of the function $\frac{P_{O2}}{P_{0}\left(T_{s}\right)+P_{O2}}exp\left(-\frac{e\varepsilon_{d}}{k_{B}T_{s}}\right)$ is obtained, i.e., we related the whole distribution to its maximum value. The calculated result is shown in Figure 9f. We can see that both curves exhibit a maximum that separates the region where the growth is suppressed by the low diffusion due to the low temperature, from the region where the growth is suppressed by the small adsorption caused by the high substrate temperature. These results seem to be generic and could be further applied for the explanation of similar growth patterns of single-crystal metal oxide nanostructures. 

Now, we can conclude the reason for the different behavior of the nanostructure growth; when at the temperature increase, CuO NWs become thick, while ZnO exhibits a large aspect ratio. According to the calculations, the oxygen adsorption energy increases with the temperature in the measured temperature range. However, for CuO the increase was more significant (Figure 9d)—from 1.25 to 1.7 eV (+0.45 eV or 36% from the initial value), while for ZnO the change was from 1.61 to 1.9 eV (+0.29 eV or 18% from the initial value). This results in the increased adsorption rates of oxygen on the side surface of the growing CuO nanostructure, and hence, the oxidation reaction is more intensive, and the nanostructure becomes thicker. On the contrary, the adsorption of oxygen on ZnO nanostructures decreases due to the increase of the temperature and not-so-strong increase of the adsorption energy, hence the nanostructure grows in height but not in diameter.

Another important point revealed by the modelling is a possible dependence of the adsorption energies on the ion flux extracted from plasma to the growth surface. Generally, the ion bombardment damages the surface and results in the generation of a large number of the surface defects, activation of chemical bonds, surface fictionalization etc. [48]. This effect can promote the formation of the nanostructures and tailor their parameters; however, the calculation of the adsorption energies is rather complicated since the methods based on a thermal equilibrium approach fail. Therefore, our calculation results may be useful concerning similar systems, where a plasma-enhanced approach is applied.

In this subsection, chemical reactions occurring for the growth of ZnO, CuO, and Fe_2_O_3_ nanostructures under the plasma-based process have been proposed. Moreover, a growth model has been developed for the interpretation of different growth behaviors of ZnO and CuO nanostructures by considering the activation energy *ε_d_* of the copper (or zinc) diffusion, and the density of the oxygen molecules adsorbed on the substrate surface.

### 3.5. Advantages of Plasma-Based Process and Outlook of This Work 

In comparison with neutral gas-based (thermal CVD) processes, the plasma-based approach has several advantages: (i) The plasma generates reactive oxygen radicals in the gas phase. This eliminates the need to heat the surface to high temperatures when dissociation of oxygen on the surface becomes effective. (ii) The plasma also creates an electric field through the formation of the plasma sheath. This electric field can effectively guide the nanowire growth and enable the growth of vertically aligned nanostructures. (iii) The plasma provides localized surface heating through recombination of as-created oxygen atoms and also through ion bombardment and neutralization, which can further increase the surface temperature. (iv) Plasma can also create localized nucleation spots via the ion bombardment as well as some other plasma-related effects, which makes the use of metal catalysts unnecessary. (v) The morphology, diameter, faceting, crystallinity, and composition of the metal oxide nanostructures can be effectively tailored by the process parameters.

Future work should involve in situ plasma diagnostic tools such as Langmuir probe and optical emission spectroscopy, to investigate the chemical reactions during the growth of metal oxide nanostructures in our custom-made plasma-enhanced horizontal tube furnace deposition system. Further comparisons of our process with other plasma-based processes, such as inductively coupled plasma process, and in-depth studies of the difference between our process and other plasma-based processes for the growth of metal oxide nanostructures are envisaged. 

## 4. Conclusions

Several metal oxide nanostructures were grown on the metal foils by the customized plasma-enhanced thermal oxidation process. The morphology and microstructure of the synthesized metal oxide nanostructures can be effectively controlled by the process temperatures. Under catalyst-free conditions, plasma-treated metal foils were activated to grow metal oxide nanostructures. The synthesized single-crystal metal oxide nanostructures were investigated and characterized by XRD, SEM, TEM, and Raman spectroscopy. Moreover, a viable growth mechanism supported by nanostructure growth modelling was proposed to interpret the obtained experimental results. This work is relevant to the development of advanced processes for the synthesis of metal oxide nanostructures.

## Figures and Tables

**Figure 1 nanomaterials-09-01405-f001:**
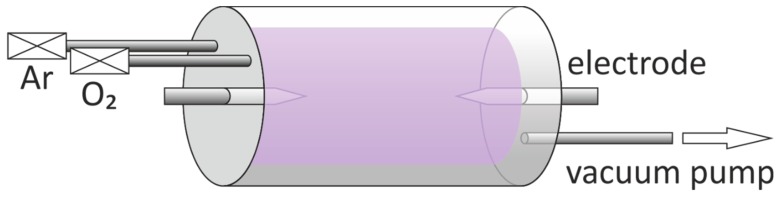
Schematic diagram of a custom-made plasma-enhanced horizontal tube furnace deposition system. The vacuum-sealed glass cylinder reactor has two electrodes with pointed ends from each side, and these two electrodes are used to ignite plasma.

**Figure 2 nanomaterials-09-01405-f002:**
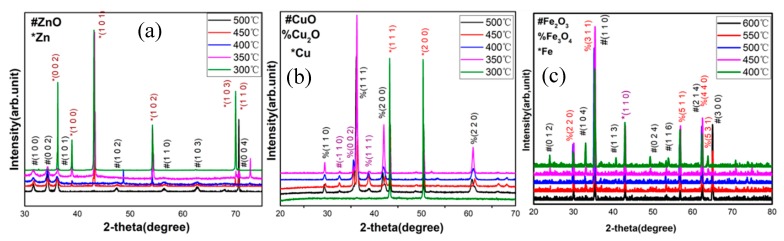
XRD patterns of the metal oxide nanostructures grown at various temperatures: (**a**) ZnO nanostructures were grown at 300–500 °C; (**b**) CuO nanostructures grown at 300–500 °C; (**c**) Fe_2_O_3_ nanostructures grown at 400–600 °C.

**Figure 3 nanomaterials-09-01405-f003:**
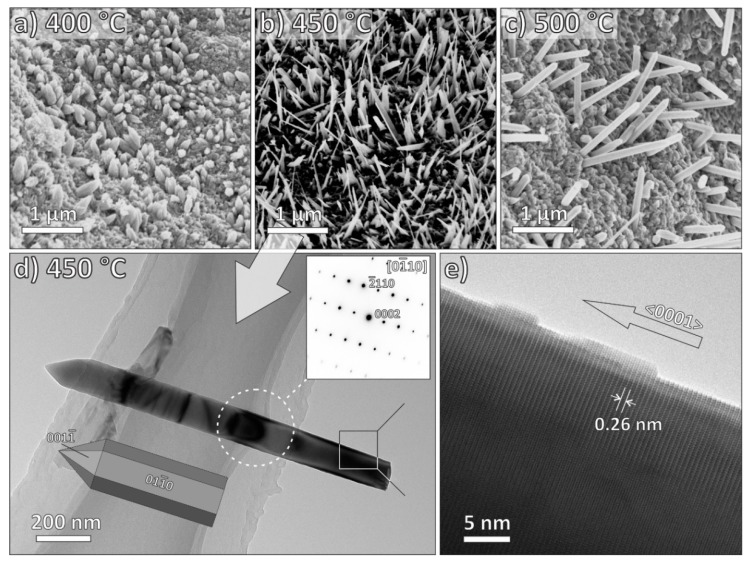
(**a**–**c**): SEM images of ZnO nanostructures synthesized at different growth temperatures: (**a**) 400 °C, (**b**) 450 °C, and (**c**) 500 °C, respectively; time period of growth was 30 min. (**d**) TEM micrograph of a single-crystalline ZnO NW grown at 450 °C with the corresponding SAED pattern (inset) and a model of crystal in the corresponding orientation. (**e**) High-resolution TEM micrograph of a ZnO nanowire, with the marked interplanar spacing of 0.26 nm. The NW is elongated in the <0001> direction.

**Figure 4 nanomaterials-09-01405-f004:**
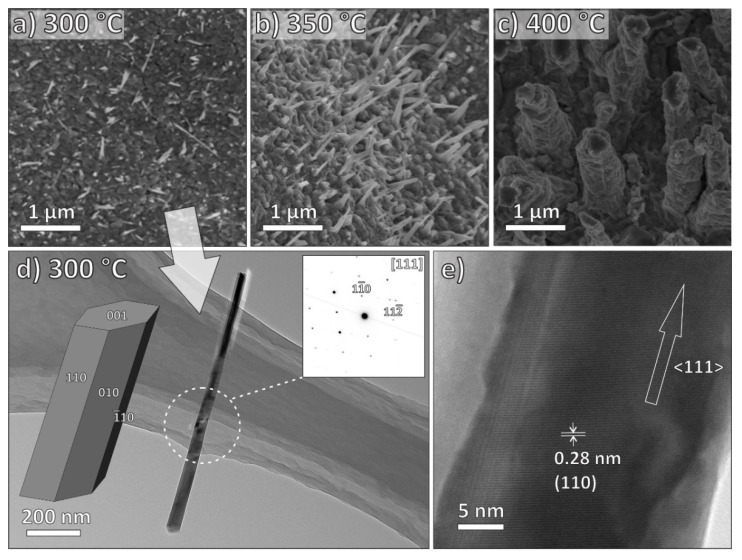
SEM images of copper oxide nanostructures synthesized at different growth temperatures: (**a**) 300 °C, (**b**) 350 °C, and (**c**) 400 °C, respectively; time period of growth was 30 min. (**d**) Top-view TEM image of a single copper oxide nanostructure grown at a processing temperature of 300 °C viewed along [111] zone axis (corresponding SAEDP in inset) and an NW model in the same orientation. (**e**) High-resolution TEM image of the copper oxide nanostructure with the marked interplanar spacing of 0.28 nm. From the SAEDP, we can obtain that the growth direction of the CuO NW is <111>.

**Figure 5 nanomaterials-09-01405-f005:**
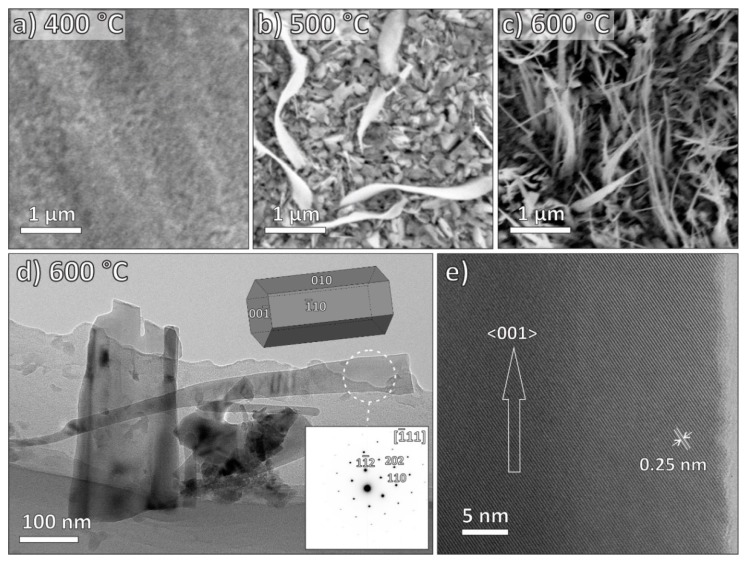
(**a**–**c**): SEM images of iron oxide nanostructures synthesized at different growth temperatures: (**a**) 400 °C, (**b**) 500 °C, and (**c**) 600 °C, respectively; time period of growth was 30 min. (**d**) TEM micrograph of a single-crystallineα-Fe_2_O_3_ nanostructure grown at 600 °C with the corresponding SAED pattern (inset) and a model of crystal in the corresponding orientation. (**e**) High-resolution TEM micrograph of anα-Fe_2_O_3_ nanostructure, with the marked interplanar spacing of 0.25 nm. The nanostructure is elongated in the <001> direction.

**Figure 6 nanomaterials-09-01405-f006:**
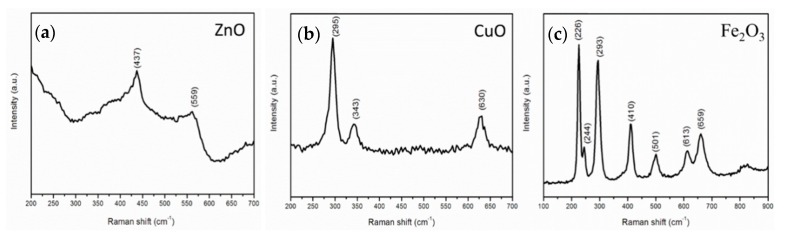
Characteristic Raman spectra of the ZnO nanostructures (**a**), CuO nanostructures (**b**), Fe_2_O_3_ nanostructures (**c**), produced at a processing temperature of 450 °C, 300 °C, 600 °C, respectively.

**Figure 7 nanomaterials-09-01405-f007:**
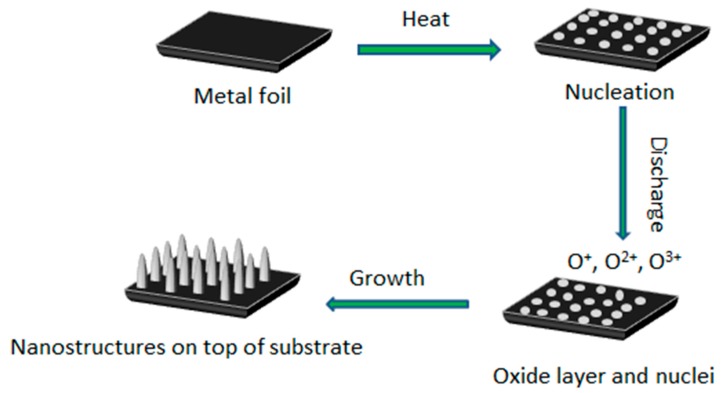
Schematic illustration of the initial stages of the growth process for the metal oxide nanostructures produced under plasma. These stages lead to the nucleation, nanowire growth, and formation of a thin oxide layer.

**Figure 8 nanomaterials-09-01405-f008:**
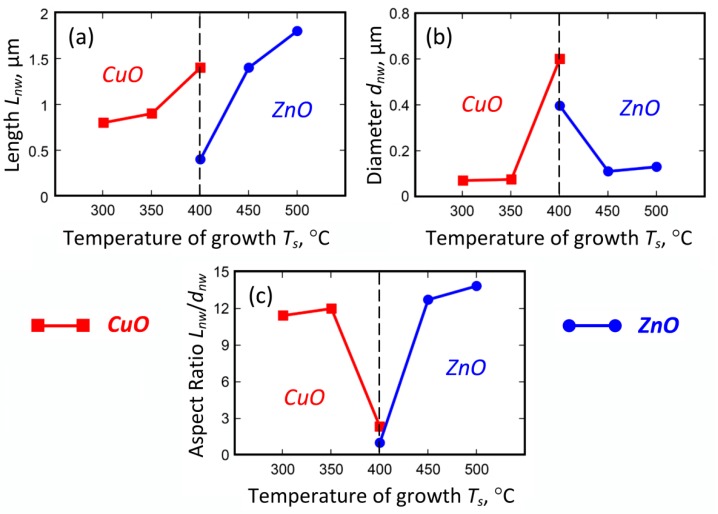
Experimental results on measurement of CuO and ZnO nanostructures: (**a**) length; (**b**) diameter; (**c**) aspect ratio.

**Figure 9 nanomaterials-09-01405-f009:**
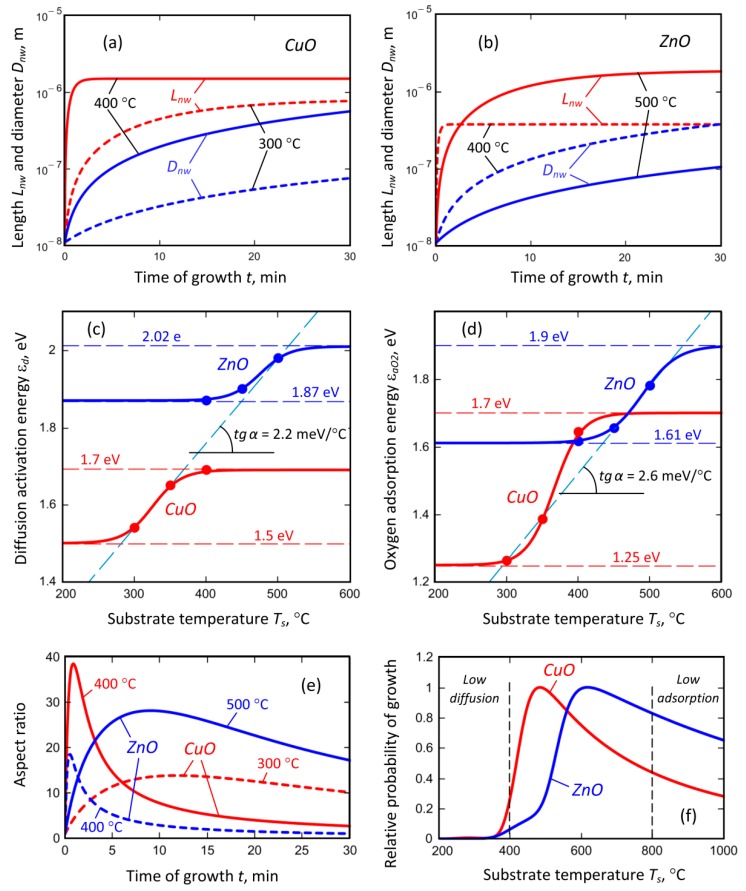
Results of calculations of the oxide dynamics: (**a**,**b**) dependencies of the nanowire length and diameter on the growth time; (**c**,**d**) dependencies of the copper diffusion activation energy and oxygen adsorption energy on the growth temperature; (**e**) nanowire aspect ratio at dependence on the time of growth; (**f**) relative probability of the nanowire growth at dependence on the substrate temperature.

**Table 1 nanomaterials-09-01405-t001:** Characteristics of the zinc oxide, copper oxide, and iron oxide nanostructures produced in this work.

MO	ZnO	CuO	Fe_2_O_3_
Growth time	30 min	30 min	30 min
Growth temperature	300–500 °C	300–500 °C	400–600 °C
Diameter	110–396 nm	70–600 nm	/
Length	0.4–1.8 μm	0.8–1.4 μm	/
Growth direction	<0001>	<111>	<001>

**Table 2 nanomaterials-09-01405-t002:** Comparison of the experimental and theoretical values for the ZnO and CuO nanostructures synthesized at different growth temperatures and a growth time of 30 min.

Metal Oxide	Growth Temperature, °C	Experimental Value		Theoretical Value	
		Length, µm	Diameter, µm	Length, µm	Diameter, µm
ZnO	400	0.4	0.396	0.374	0.377
	450	1.4	0.110	1.267	0.106
	500	1.8	0.130	1.823	0.106
CuO	300	0.8	0.070	0.763	0.075
	350	0.9	0.075	0.873	0.075
	400	1.4	0.6	1.486	0.558

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
