# Peer review of "Single-Crystalline Metal Oxide Nanostructures Synthesized by Plasma-Enhanced Thermal Oxidation"

_nanomaterials, 2019, doi:10.3390/nano9101405_

Round 1
Reviewer 1 Report
The manuscript has been significantly improved and the authors have successfully addressed my comments. Therefore I propose that the manuscript is published after some minor check of some typos spotted in the document.
Author Response
Thank you very much for this positive comment. We have carefully checked the manuscript and some typos have been revised. We have highlighted any revisions using the "Track Changes" function in Microsoft Word.

Reviewer 2 Report
This is vastly improved.
Some care still needed in the text e.g.:
Fe2O3exhibitscatalytic , magnetic
Fe2O3is a transition metal oxide
photothermalor photoconductive materials
(These could be a typesetting issues though).
"4c), the obtained copper oxide nanostructures become s turdy"?
"turdy (comparative more turdy, superlative most turdy) (informal) Resembling a piece of excrement"
Did the author mean "sturdy"?
Author Response
Thank you very much for this positive comment. We have carefully checked the manuscript and some typos have been revised. For example, “Fe2O3is a transition metal oxide” has been changed to “Fe2O3 is a transition metal oxide”; “Fe2O3exhibitscatalytic, magnetic” has been changed to “Fe2O3 exhibits catalytic, magnetic”; “photothermalor photoconductive materials” has been changed to “photothermal photoconductive materials”; “turdy” has been changed to “sturdy”……All the revisions have been highlighted using the "Track Changes" function in Microsoft Word.

This manuscript is a resubmission of an earlier submission. The following is a list of the peer review reports and author responses from that submission.
Round 1
Reviewer 1 Report
This is an interesting paper that would no doubt have many more citations if a little more care were taken in its presentation. It would be beneficial to include how your results compare to others, and should theybe unique then perhaps an outline of future work. Some more care is needed with the English too. "had two pointy electrodes" should be replaced with "had two electrodes with pointed ends" or something even more suitable. Other examples exist in the text that could be improved through the aid of a native speaker.
Reviewer 2 Report
This work presents an effort to characterise and model the growth mechanism of Zn, Cu, and Fe oxides, in an effort to unlock the mechanism of growth using plasma-enhanced thermal oxidation.
This is a very interesting piece of research which might lead to a deep understanding on the mechanism governing growth in plasma thermal oxidation.
However there are some issues that need to be addressed in order of the publication to be of adequate quality to be published in Nanomaterials. These are the following:
The title proposed a generic solution on the mechanism. On the contrary only 2 oxides are investigated (and one more is characterised in terms of Raman, SEM, XRD).
The manuscript needs English revision in some parts the reader cannot understand what the authors claim.
Referencing needs to be revised. For example "The template method can assure good control over the morphology of nanostructures, but it is expensive and time-consuming. Hydrothermal synthesis is fast, efficient and relatively cheap, but the morphology of the product is only poorly controllable by, e.g. surfactants. Laser ablation synthesis can provide more control over the morphology and quality of nanomaterials, but the process itself is
expensive and unsuitable for mass production." needs references. There are several other parts in the introduction that need to be referenced properly.
SEM images shown are after growth of nanostructures. What is the time period of growth shown in the images?
SEM images seem tilted. What is the tilt angle? Please provide more information on that.
Show a larger view of the surface with nanostructures not only the zoomed view for the reader to understand how the structures grow on the substrate.
Add scale bars on all SEM images.
In section 3.2 you define lengths and diameters. In order for this to be claimed you need a statistical analysis with a significant sample. How many images did you use in order to define these sizes? What method did you use to extract these numbers? If you did not, please do and provide the outcomes.
You might consider to compare the 3 different oxides. An image and/or table will help the reader to understand what is happening.
Could sonication be avoided to avoid breakage and thus the full length of the structures be characterised?
I do not understand why the authors did not use the same temperatures in all of the oxides, and even more why the gradient of changes shown in the SEM images are not the same. This is very inconsistent and does not aid in the direct comparison of the 3 cases.
It would be very helpful if the authors summarised the findings on each section especially in the experimental part. It is difficult for the reader to follow the logic of the authors. the way it is currently written.
English in growth mechanism section is not adequate and should be revised. Please rewrite the section and especially the part with the microhillocs.
I fail to understand the logic behind why only ZnO and CuO are investigated in the growth mechanism section. Could you provide a reasoning and an argument why Fe2O3 is not.
In page 20 experimental values are compared with theoretically predicted ones. Please make a table to understand better this comparison. A number is missing for either of the two, but still the authors claim there is an agreement between their values.
"It should be stressed... ion bombardment of the surface". How do you propose to cope with this anomaly you have identified?
You can use tables to show and compare the three named oxides studied in this paper.